# RASPOTION—A New Global PET Dataset by Means of Remote Monthly Temperature Data and Parametric Modelling

**Aristoteles Tegos** [1,*], **Nikolaos Malamos** [2] **and Demetris Koutsoyiannis** [1]

1 Department of Water Resources and Environmental Engineering, National Technical University of Athens, Heroon Polytechneiou 5, GR-157 80 Athens, Greece; dk@itia.ntua.gr
2 Department of Agriculture, University of Patras, GR-272 00 Amaliada, Greece; nmalamos@upatras.gr
* Correspondence: tegosaris@yahoo.gr

**Abstract:** Regional estimations of Potential Evapotranspiration (PET) are of key interest for a number of geosciences, particularly those that are water-related (hydrology, agrometeorology). Therefore, several models have been developed for the consistent quantification of different time scales (hourly, daily, monthly, annual). During the last few decades, remote sensing techniques have continued to grow rapidly with the simultaneous development of new local and regional evapotranspiration datasets. Here, we develop a novel set T maps over the globe, namely RASPOTION, for the period 2003 to 2016, by integrating: (a) mean climatic data at 4088 stations, extracted by the FAO-CLIMWAT database; (b) mean monthly PET estimates by the Penman–Monteith method, at the aforementioned locations; (c) mean monthly PET estimates by a recently proposed parametric model, calibrated against local Penman–Monteith data; (d) spatially interpolated parameters of the Parametric PET model over the globe, using the Inverse Distance Weighting technique; and (e) remote sensing mean monthly air temperature data. The RASPOTION dataset was validated with in situ samples (USA, Germany, Spain, Ireland, Greece, Australia, China) and by using a spatial Penman–Monteith estimates in England. The results in both cases are satisfactory. The main objective is to demonstrate the practical usefulness of these PET map products across different research disciplines and spatiotemporal scales, towards assisting decision making for both short- and long-term hydro-climatic policy actions.

**Keywords:** RASPOTION; potential evapotranspiration; parametric model; remote sensing; hydrological calibration





## 1. Introduction

Evapotranspiration (ET) is a crucial element of the hydrological cycle relevant in a wide range of geosciences, since it represents the combined water losses from soil surface and vegetation. It is influenced by several meteorological variables such as air temperature, solar radiation, wind speed, and relative humidity. The literature proposes several approaches to quantify the process in terms of actual evapotranspiration, potential evapotranspiration (PET) or reference evapotranspiration. By definition, PET refers to "the rate at which evapotranspiration would occur from a large area completely and uniformly covered with growing vegetation, which has access to an unlimited supply of soil water, and without advection or heating effects" [1]. PET is different from the actual evapotranspiration, which also depends on the actual soil water supply, mainly driven by the precipitation regime. In recent decades, advanced methods have been introduced for ET and PET estimation, the most recent being the remote sensing technique, incorporating aerial and satellite imagery [2–5]. Generally, the classification of remote sensing for ET assessment includes four groups referred to as empirical, direct, residual, inference and deterministic models [6]. The most well-known approach for the actual evapotranspiration estimation for daily and monthly time step is the modified surface energy balance algorithm for land (SEBAL) model [7]. A limited number of studies have focused on the global PET assessment utilizing

remote sensing tools. Specifically, the global distribution of potential evaporation has been calculated from the Penman–Monteith equation using satellite and assimilated data for a 24-month period, i.e., January 1987 to December 1988 [8].

The Parametric model is a temperature-based model that requires only temperature data and utilizes a parsimonious expression for the potential evapotranspiration (PET) estimation [9]. It replaces some of the variables and constants that are used in the standard Penman–Monteith model by regionally varying parameters, which are estimated through calibration [10–12]. The large-scale Parametric model application was satisfactory, and it outperformed the efficiency of several simplified models such as Hargreaves, Thornthwaite, Oudin, and Jensen–Haise.

In this study, a new global PET monthly dataset is introduced by applying the Parametric model using the remote sensing data (LANDSAT) of mean air temperature, provided by a recent remote mean temperature dataset from 2003 to 2016. As most global applications refer to the actual evapotranspiration assessment [2–5], this dataset may contribute to hydrological balance modelling and agrometeorological applications.

## 2. Materials and Methods

The Parametric model employs physically consistent parameters distributed over the globe, overcoming the main weakness of the Penman–Monteith model, which is the necessity of simultaneous observations of four meteorological variables [10–14].

The modified Parametric model implements two instead of three parameters, namely parameter $a'$ in the numerator and parameter $c'$ in the denominator of the formula:

$$\text{PET} = \frac{a' R_a}{1 - c' T} \tag{1}$$

where PET is the potential evapotranspiration (mm), $R_a$ (kJ m$^{-2}$) is the extra-terrestrial radiation, $a'$ (kg kJ$^{-1}$), and $c'$ ($°$C$^{-1}$) are the calibrated parameters and $T$ ($°$C) is the monthly mean air temperature. As already stated in Tegos et al. [12], the parameters have some physical correspondence to the Penman–Monteith equation, since the product $a' R_a$ represents the overall energy term (i.e., incoming minus outgoing solar radiation), while the quantity $1 - c' T$ approximates the denominator term of the Penman–Monteith formula. More information about the parameters $a'$ and $c'$ and their spatial patterns across the globe can be found in [12].

The model was applied globally using the values of parameters $a'$ and $c'$ at the locations of 4088 stations of the FAO-CLIMWAT database, which presented positive efficiency according to the Nash–Sutcliffe criterion during calibration (Figure 1). These values were interpolated over the globe using the inverse distance weighting (IDW) technique into a geographical information system (GIS) [12].

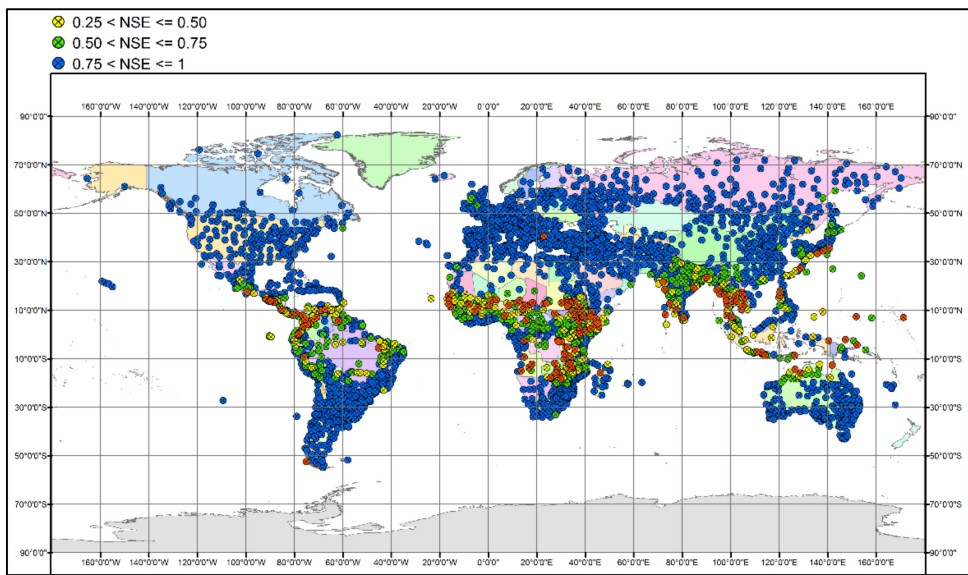

**Figure 1.** CLIMWAT meteorological stations network and distribution of efficiency (from [12]).

The extra-terrestrial radiation ($R_a$) monthly raster datasets were derived from the respective daily values using an analytical mathematical expression [12], from $-90°$ to $+90°$ of latitude with a step of $0.05°$ for normal and leap years (Figure 2), taking into consideration the polar daylight and polar night periods [15].

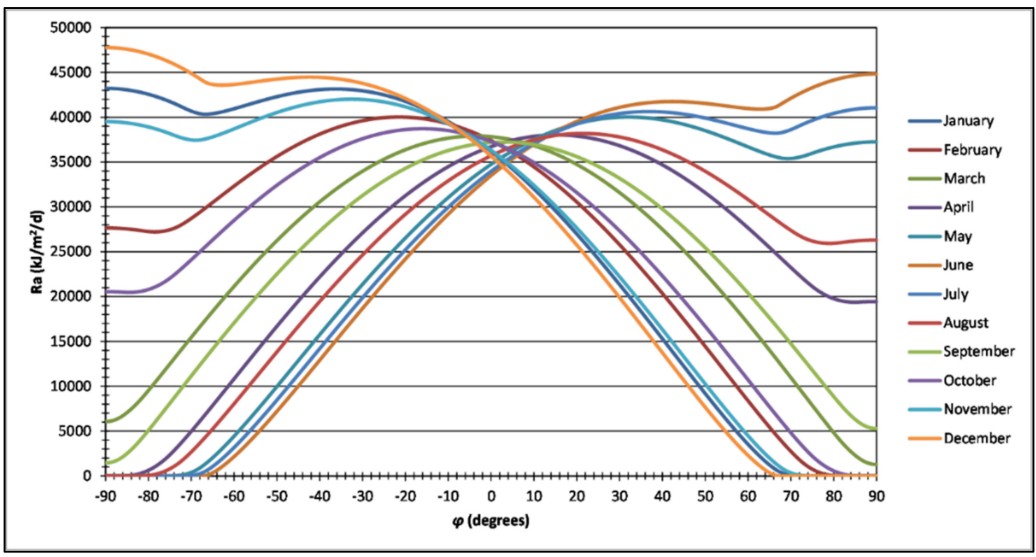

**Figure 2.** Mean monthly extra-terrestrial radiation ($R_a$) for latitudes $-90°$ to $+90°$.

The mean air temperature values, covering a period from 2003 to 2016, were acquired as raster datasets from the recent analysis presented in [16]. Figure 3 shows the spatial temperature variation across the globe in June 2011.

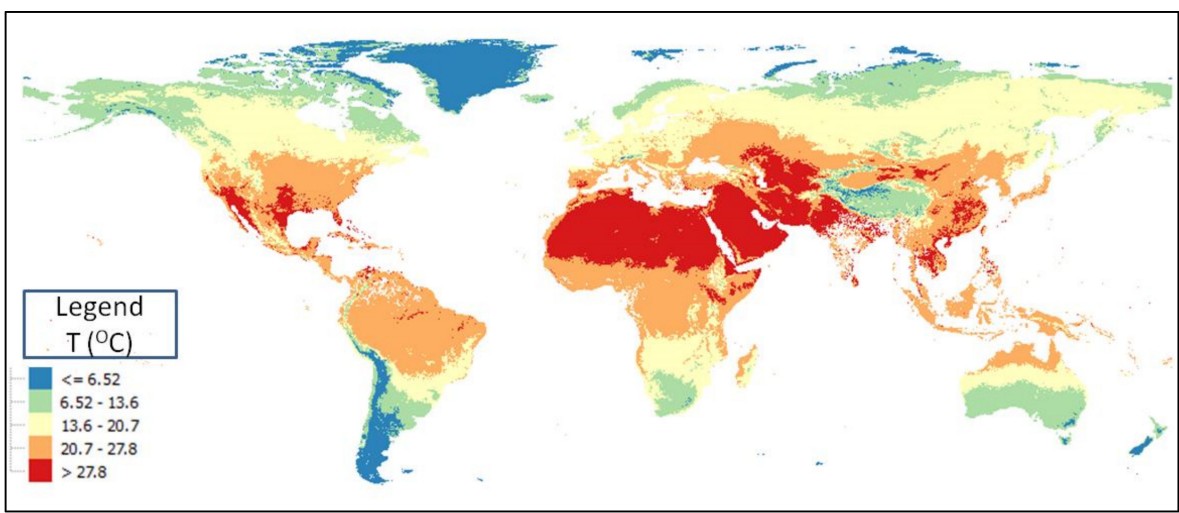

**Figure 3.** Monthly air temperature in June 2011.

All three layers of information were embedded in GIS and constituted a framework that permitted quality control screening by application of reasonable thresholds to exclude the outliers in the PET values. The maps obtained using the parametric method were produced in a GIS environment by applying Equation (1) with the required raster datasets, i.e., parameters $a'$ and $c'$, extra-terrestrial radiation $R_a$, and monthly mean air temperature $T$.

## 3. Results

### 3.1. PET Global Mapping

Following the above presented methodology, a monthly PET global dataset was acquired covering the period 2003–2016. Figure 4, visualizes the PET distribution for a representative month (August 2011) globally.

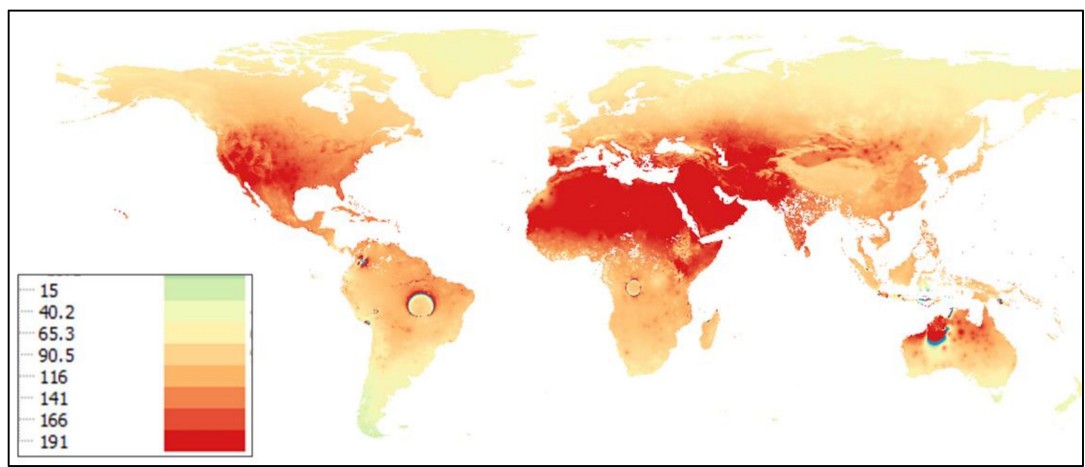

**Figure 4.** Global PET map (mm/month) for August 2011.

In Eurasia, where PET monthly values range from 19 to 239 mm, PET increases from north to south. The latter is well explained from the similar variation of temperature and extra-terrestrial radiation. The highest values were observed in the Middle East, where extremely arid climatic conditions occur. A pattern similar to Eurasia was observed in North America, with the highest values at regions near the equator (e.g., Mexico) and lowest in Canada, Alaska and Greenland. In South America, PET decreases from north to south. Some inconsistencies in the area of Amazon and some peculiarly low values in the area of equator can be explained from the limitations of the Parametric model to represent

the combined effect on PET estimation of relative humidity and wind speed, which are key drivers of the evapotranspiration processes across these areas, influencing the net incoming solar radiation and the evaporation demand, as detailed in Tegos et al. [12].

High monthly PET values were acquired in the equatorial zone for Africa, mainly in the lower Congo, where the hydro-meteorological observations were limited in the Parametric model calibration. The decreasing trend from north to south in Oceania follows the pattern of radiation and air temperature variation.

### 3.2. Validation

The RASPOTION remote sensing dataset has been compared against the Penman–Monteith timeseries, estimated at 26 meteorological stations across the globe, listed in Table 1. The validation was carried out for long-term monthly Penman–Monteith samples by examining the coefficient of efficiency (CE) due to Nash and Sutcliffe [17] in different countries, namely, the USA, Germany, Spain, Ireland, Greece, Australia, and China, and with different climatic regimes. Penman–Monteith timeseries were retrieved by different databases such as CIMIS network (https://cimis.water.ca.gov/), European Climate Assessment data set (http://eca.knmi.nl/), Australian Bureau of Meteorology (http://www.bom.gov.au/watl/eto/, accessed on 1 February 2022), the Irish Meteorological Service—Met Éireann (https://www.met.ie/)—and a previously published paper [18]. The new RASPOTION dataset shows an excellent performance across different climatic regimes. Only two stations (Shanxi, Sydney Airport) demonstrate a moderate performance; however, the coefficient of efficiency was above a threshold that can safely allow its further operational use.

**Table 1.** Validation dataset.

| Station | Country | Period | CE |
|---|---|---|---|
| Kostakioi | Greece | 04/2008–07/2013 | 89 |
| Mace Head | Ireland | 10/2010–11/2016 | 90.9 |
| Zaragoza | Spain | 01/2003–11/2009 | 92.8 |
| Alicante | Spain | 01/2003–10/2009 | 92.4 |
| Munchen | Germany | 01/2003–06/2013 | 90.0 |
| Karshue | Germany | 01/2003–08/2009 | 89.2 |
| Hamburg | Germany | 01/2003–06/2013 | 93.7 |
| Frankfurt | Germany | 01/2003–06/2013 | 96.7 |
| Dusseldorf | Germany | 01/2003–06/2013 | 94.7 |
| Dresden | Germany | 01/2003–06/2013 | 92.9 |
| Bremen | Germany | 01/2003–06/2013 | 94.9 |
| Angermunde | Germany | 01/2003–06/2013 | 95.2 |
| Aachen | Germany | 01/2003–11/2005 | 91.5 |
| Tulelake | USA | 01/2003–11/2005 | 79.2 |
| Meloland | USA | 01/2003–06/2013 | 89.0 |
| Manteca | USA | 01/2003–06/2013 | 93.1 |
| Temecula | USA | 01/2003–06/2013 | 84.5 |
| Buntigville | USA | 01/2003–06/2013 | 89.6 |
| Mc Arthur | USA | 01/2003–06/2013 | 89.5 |
| Davis | USA | 01/2003–06/2013 | 93.1 |
| Tunnack Firestation | Australia | 01/2009–12/2014 | 90.5 |
| Adelaide airport | Australia | 01/2009–12/2014 | 83.2 |
| Sydney Airport | Australia | 01/2009–12/2014 | 43.2 |
| Alice Springs | Australia | 01/2009–12/2014 | 84.1 |
| Albany airport | Australia | 01/2009–12/2014 | 90.6 |
| Shanxi | China | 01/2003–12/2014 | 22.3 |

Following RASPOTION's evaluation performance at a point basis across the globe, an advanced comparison is presented here that demonstrates its performance against spatial PET monthly maps provided by the Environmental Agency throughout England. The latter

dataset has been developed using the daily FAO-56 PET method in an extensive gauge meteorological network across England, and spatial mapping with the use IDW method. A free data access is provided by DEFRA Data Services Platform. Figure 5 shows the PET monthly variation across England. PET ranges from 25 mm/month to 135 mm/month and increases from North to Southeast.

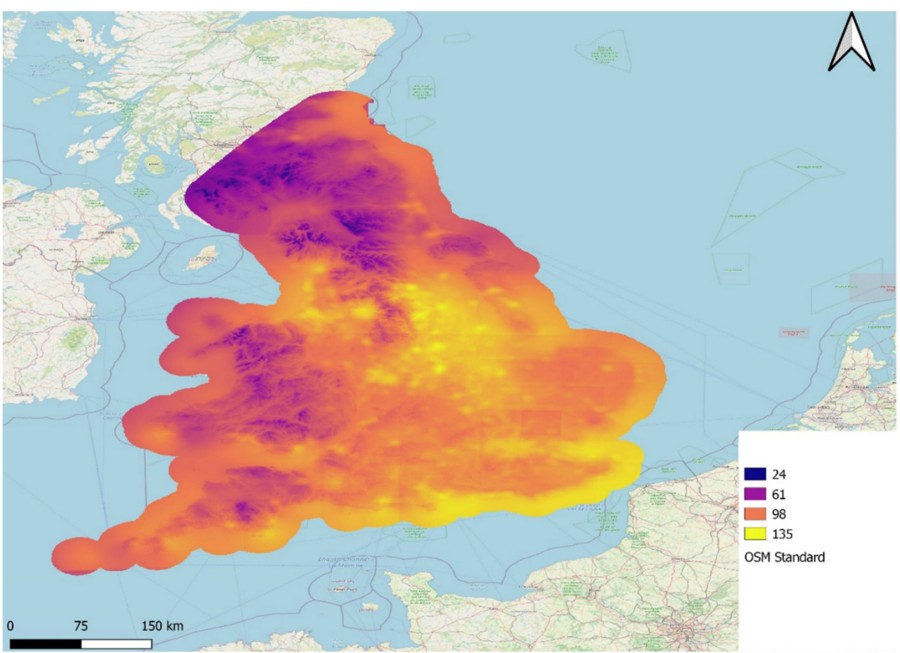

**Figure 5.** PET monthly map (DEFRA- June 2011 PET in mm/month).

By comparing RASPOTION's June 2011 raster map with the DEFRA PET map a very good performance is achieved, as the actual difference for the majority of England (around 80%) is up to ±9 mm/month (Figure 6), corresponding to a range of up to 12% of overestimation and 9% underestimation of actual PET, respectively (Figure 7).

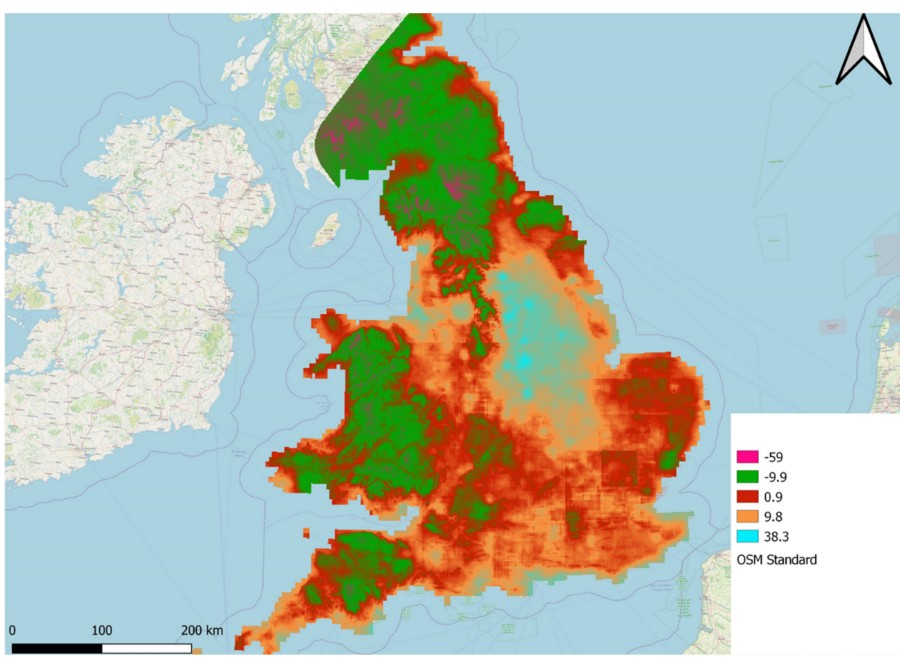

**Figure 6.** Spatial difference RASPOTION against DEFRA PET (mm/month).

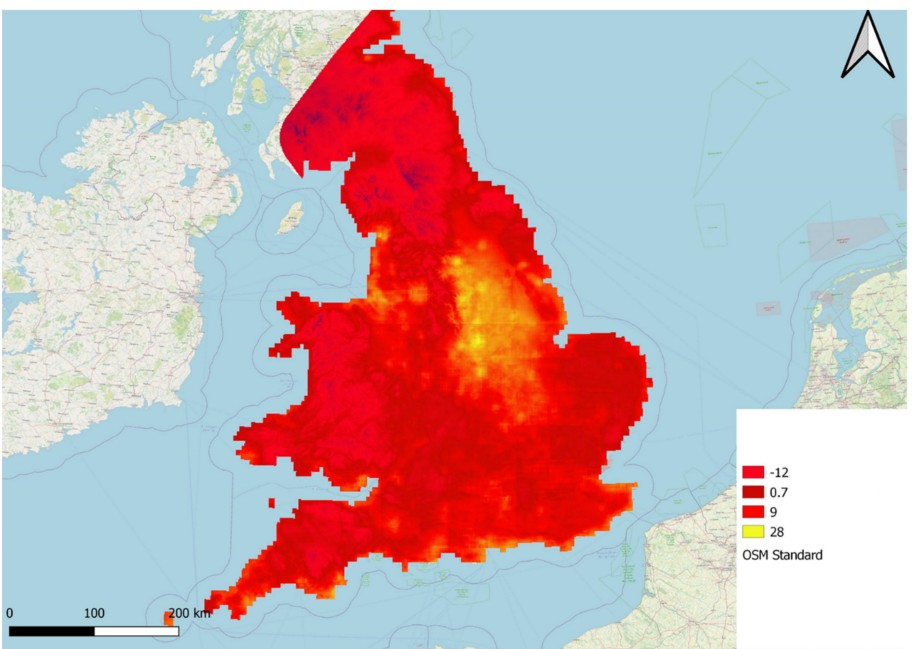

**Figure 7.** Ratio difference RASPOTION against DEFRA PET.

## 4. Discussion

In previous studies, extended discussion has shown that the Parametric PET framework provides satisfactory results with some limitations [12], also outlined here. In some areas with high values of relative humidity and wind speed, the existing PET parametric approach fails to efficiently reproduce the PET characteristics and further improvements to the parametric approach are recommended locally. Those areas are located in sub-Saharan Africa (Kongo, Zimbabwe, Rwanda and Uganda) and South America, adjacent to the Amazon River catchment (Brazil). It should be noted that the FAO-CLIMWAT data were scarce and poor in Sub-Saharan and South American territories. In these territories the calibrated parameters $a'$ and $c'$ are indicative and we recommend further calibration should take place. The missing hydrometeorological variables and the lack of calibration was also shown in a recent study by dos Santos et al. [19], where the parametric model displayed a moderate performance in subtropical areas. Long-term measurements of relative humidity, wind speed, temperature and radiation in conjunction with a new calibration of a revised parametric formula would lead to substantial improvements in the existing parametric estimates. The work should be accompanied and supported by a thorough statistical investigation between the climatic factors (i.e., radiation, wind speed, relative humidity) to identify the dependence of PET with any other variable except temperature.

Nonetheless, hydrologists, agronomists and other scientists with potential interest in this dataset could make efficient use of it, in about 80% of the earth's territory based on our previous studies [12]. Taking into account the fact that other global PET datasets are not available, the potential benefits of our new dataset may include the following:

(1) In the applications of physically based hydrological models which use PET as an input in catchment modelling. This acquires higher usefulness as we are moving forward toward global scale hydrological models. It is already highlighted by several researchers that the use of accurate PET estimates is of great importance for the reproduction of physically based hydrological responses [20] and its use in calibrating complex physically based hydrological models [21,22].

(2) In the crop-water demand assessment. The integration of the monthly PET and the cropping pattern quantifies the monthly water needs according to vegetation. Accurate PET estimates with fewer demands of meteorological data, combined with

modern techniques such as the use of a drone for micro-farming data gathering [23] greatly support the food–water nexus.

(3)   In climatological studies and drought assessment. The use of reliable PET models greatly influences several well-known indexes such as Palmer Drought Severity Index [24], aridity index [9] and Surface Wetness Index [25].

(4)   In building and evaluating environmental resilience indexes for developing and defining new multi-dimensional approaches. Such approaches would ensure an accurate decision-making basis for sustainable ecosystem management [26].

## 5. Conclusions

As part of the PET Parametric model, a new global PET monthly dataset based on remote-sensing temperature data is introduced, covering the period 2003–2016. This global dataset was produced using the Parametric formula which uses as input variables extra-terrestrial radiation and mean air temperature. The remote temperature data have been taken from a freely available dataset provided by Hooker et al. [16]. Previous analyses with this approach showed satisfying performance through validation under several climatic regimes and different validation procedures. In regions where the available hydro-meteorological information was scarce or insufficient, the modelling results were weak in terms of PET's physical interpretation. In these areas the RASPO-TION dataset should be used with caution. The dataset is open and freely accessible from http://www.itia.ntua.gr/2167/ (accessed on 1 February 2022), where a total number of 168 monthly global raster files (GeoTIFF) are stored. Overall, for the majority of the Earth's surface, a reliable monthly PET dataset is compiled and made available to scientists across different research disciplines in order to assist scientific studies into the global hydrological cycle and decisions for both short- and long-term hydro-climatic policy actions. Future research, consisting of exploration of model parameters and their clustering across the globe, will provide area specific parameters, thereby excluding the need for local calibration and further enabling the use of the Parametric framework.

**Author Contributions:** A.T.; methodology, data mining, draft reporting N.M.; draft reviewing $R_a$ modelling, D.K.; founding the parametric method, draft reviewing/editing, supervision. All authors have read and agreed to the published version of the manuscript.

**Funding:** This research received no external funding.

**Data Availability Statement:** The dataset is open and freely accessible from http://www.itia.ntua.gr/2167/ (accessed on 1 February 2022), where a total number of 168 monthly global raster files (GeoTIFF) are stored.

**Acknowledgments:** The manuscript is an invited paper as part of the Special Issue "Advances in Evaporation and Evaporative Demand" organized by Hydrology journal. We are grateful to Scientific Editor for handling it and Assistant Editor for inviting us to submit our work. We are also thankful to two anonymous reviewers for the constructive comments which helped us to improve our manuscript substantially. We, finally, thank Brendan Larkin for his final English proofreading and suggested corrections.

**Conflicts of Interest:** The authors declare no conflict of interest.

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
