# Peer review of "RASPOTION—A New Global PET Dataset by Means of Remote Monthly Temperature Data and Parametric Modelling"

_hydrology, doi:10.3390/hydrology9020032_

Round 1
Reviewer 1 Report
The manuscript is a logical continuation of results of the paper of Tegos et al. “Parametric Modeling of Potential Evapotranspiration: A Global Survey.” Water 2017, 9, 795. In this work, the authors carried out a detailed study to substantiate the parametric formula for estimating potential evaporation, calibrated the parameters of this formula according to the data of 4300 meteorological stations, and validated the model for a number of stations. In the manuscript, these results of the work were supplemented by the authors with global data sets of remote sensing of air temperature and maps of potential evaporation on a global scale were obtained using GIS technologies. These maps represent the main important scientific and applied results of the work.
There is the following comment to the paper. In Conclusion, the authors recommend the obtained monthly PET data sets for scientific and applied research of the global hydrological cycle, but there is no link to the Internet resource where these data are presented. It is necessary either to provide this link, or to explain how these data can be obtained.
The text of manuscript is well structured and sufficiently illustrated with graphic and tabular materials. There are only two minor misspells.
- Page 3. Figure 3 shows the spatial temperature variation across the globe in June 2011.
Figure 3. Monthly air temperature in June 2010. The year is wrong.
- Page 5 Table 1. What is CE (obviously the coefficient of efficiency)? It is necessary to determine.
References to own papers of the authors are appropriate and not overloaded.
The manuscript is recommended for publication after minor corrections.
Author Response
We thank for comments provided. A detailed response is attached

Reviewer 2 Report
- How appropriate was the use of IDW for interpolation, globally? What other methods were considered for the interpolation of the parametric model parameters?
- What is the time period for the dataset (e.g., it is mentioned that 4088 stations were used) -> 2003 – 2016 – include this in the abstract
- What are some limitations of the global dataset (e.g., is there bias regarding model performance by location across the globe)?
- What is unique about RASPOTION relative to other potentially available and comparable datasets?
- The abstract mentions that the main objective of the essay is a demonstration of the dataset, but is that not just another objective in addition to introducing RASPOTION?
- The Introduction includes a rather limited amount of cited literature (e.g., line 43 cites no literature at the end of the sentence: “In the last decades advanced methods were introduced for the ET and PET estimation with the most recent being the remote sensing techniques incorporating aerial and satellite imagery.”; and here: “A limited number of studies have focused on the global PET assessment utilizing remote sensing tools”; and here: “The Parametric model is a radiation-based model that requires only temperature data and utilizes a parsimonious expression for the potential evapotranspiration (PET) estima-tion”).
- What does satisfactory mean (line 56)? Was it satisfactory everywhere?
- I encourage some brief but additional presentation about the parameters a’ and c’ rather than simply citing an article (line 73/74).
- June 2010 or 2011 – figure 3
- Are the anomalies present in Figure 4 in S. America, Africa, and Australia discussed? Where do they come from?
- What is uniquely presented in this essay relative to other related publications?
Author Response

(The authors gave the same response as above.)

Round 2
Reviewer 2 Report
I suggest final pass throughs to correct English grammar and spelling mistakes before publication. I don't want to review the manuscript again. Thank you.
Author Response
Dear reviewer,
Thank you for your comments on both review rounds. The paper has now been checked by native English speaker and some minor changes have been made.
